# Forest management in southern China generates short term extensive carbon sequestration

Xiaowei Tong[1,2,13], Martin Brandt [2,13], Yuemin Yue[1,3]*, Philippe Ciais [4], Martin Rudbeck Jepsen[2], Josep Penuelas [5,6], Jean-Pierre Wigneron [7], Xiangming Xiao [8], Xiao-Peng Song [9], Stephanie Horion[2], Kjeld Rasmussen[2], Sassan Saatchi[10], Lei Fan [7], Kelin Wang[1,3]*, Bing Zhang[11], Zhengchao Chen[11], Yuhang Wang [12], Xiaojun Li[7] & Rasmus Fensholt [2]

Land use policies have turned southern China into one of the most intensively managed forest regions in the world, with actions maximizing forest cover on soils with marginal agricultural potential while concurrently increasing livelihoods and mitigating climate change. Based on satellite observations, here we show that diverse land use changes in southern China have increased standing aboveground carbon stocks by $0.11 \pm 0.05$ Pg C $y^{-1}$ during 2002–2017. Most of this regional carbon sink was contributed by newly established forests (32%), while forests already existing contributed 24%. Forest growth in harvested forest areas contributed 16% and non-forest areas contributed 28% to the carbon sink, while timber harvest was tripled. Soil moisture declined significantly in 8% of the area. We demonstrate that land management in southern China has been removing an amount of carbon equivalent to 33% of regional fossil $CO_2$ emissions during the last 6 years, but forest growth saturation, land competition for food production and soil-water depletion challenge the longevity of this carbon sink service.

[1] Key Laboratory for Agro-ecological Processes in Subtropical Region, Institute of Subtropical Agriculture, Chinese Academy of Sciences, Changsha, China. [2] Department of Geosciences and Natural Resource Management, University of Copenhagen, Copenhagen, Denmark. [3] Huanjiang Observation and Research Station for Karst Ecosystem, Chinese Academy of Sciences, Huanjiang, China. [4] Laboratoire des Sciences du Climat et de l'Environnement, CEA-CNRS-UVSQ, CE Orme des Merisiers, Gif sur Yvette, France. [5] CSIC, Global Ecology Unit CREAF-CSIC-UAB, Bellaterra, Spain. [6] CREAF, Cerdanyola del Vallès, Spain. [7] ISPA, UMR 1391, INRA Nouvelle-Aquitaine, Bordeaux Villenave d'Ornon, France. [8] Department of Microbiology and Plant Biology, University of Oklahoma, Norman, OK, USA. [9] Department of Geosciences, Texas Tech University, Lubbock, TX 79409, USA. [10] Jet Propulsion Laboratory, California Institute of Technology, Pasadena, CA, USA. [11] Institute of Remote Sensing and Digital Earth, Chinese Academy of Sciences, Beijing, China. [12] State Key Laboratory of Earth Surface Processes and Resource Ecology, Faculty of Geographical Science, Beijing Normal University, Beijing, China. [13] These author contributed equally: Xiaowei Tong, Martin Brandt. *email: ymyue@isa.ac.cn; kelin@isa.ac.cn

Mitigating climate change while securing livelihoods is a major societal challenge in the twenty-first century. Reducing emissions of $CO_2$ from the combustion of fossil fuels and deforestation, and developing land-based technologies to sequester atmospheric $CO_2$ are crucial to limit global warming[1]. Current research indicates that vegetation growth and photosynthetic activity are increasing globally[2,3], which translates into an increased terrestrial biomass carbon sink and thereby contributes to mitigate the growth rate of atmospheric $CO_2$[4]. Cropland and pasture land may accumulate carbon in soils[2,5], but the storage potential is uncertain, whereas many studies indicate that adequate forest management can enhance biomass carbon stocks[6,7].

Forestation (reforestation and afforestation) sequesters carbon in biomass, but the scalability of this land management option for meeting ambitious warming limitation targets has been questioned due to the sheer amount of land area required[1]. Further, forestation of arable land results into trade-offs with the local food production[7,8], and forests tend to have a high water use, so that new plantations decrease soil moisture (SM) and renewable freshwater resources[9,10].

Moderate forest harvesting practices on forested farmlands can generate carbon stocks and at the same time provide economic output from timber products[11]. As such, expansion of managed forests provides alternative income for farmers, and an increase in the time-averaged standing carbon stock. Most of our current knowledge of the impact of forest management on climate change, however, is based on models and site-scale evidence, and real-world empirical information about the performance of diverse forest management strategies in relation to carbon storage and SM at larger observational scale is needed[12].

Southern China is an important case region in the sub-tropics, as it hosts a large number of carbon-emitting economic activities and because it experienced intensive land use changes, being now among the most dynamic areas of managed forests in the world[13–17]. Since 2000, government-funded efforts to combat land degradation and fight poverty have been put in place[18,19]. The main land use decisions have been to expand the forested area and reduce agriculture on marginal sloping lands, and to intensify crop cultivation on more fertile/less erodible soils. Planted trees have generated income[19] and woody vegetation cover has greatly increased[18,20]. The results of this land use transformation are relevant to other subtropical regions that are experiencing fast tree-cover loss and continued expansion of croplands in hill slopes[21,22].

Management practices for forested land in southern China include protection of existing (old-growth) forests, recovery of deforested areas, afforestation of croplands for conservation purpose, and development of industrial timber and paper production[19]. These diverse actions produce contrasted forest types with different carbon sequestration and storage potentials, which are evaluated in this study. We apply a range of observational data based on optical satellite imagery (including a full area coverage of southern China at 2 m) and low-frequency passive microwaves to identify different management strategies and biomass carbon sinks from 2002 to 2017 (see Methods; refs. [23–25]). Low-frequency microwave data are also used to evaluate SM trends during the period 2010–2017 (see Methods; ref. [24]). Results show that new forests are widespread over southern China representing an extensive carbon sink and offsetting an amount of carbon equivalent to one-third of the regional fossil $CO_2$ emissions, but possible soil water depletion and the limited availability of arable land challenge the longevity of the carbon sink service.

## Results

### Tree cover and $CO_2$ emissions increased between 2002 and 2017.

Forest cover has increased considerably at the expense of bare ground and short vegetation (grassland and cropland) in southern China (we studied eight provinces covering China's largest forests[17,26]) from an average tree cover of 21% in 1982 to 27% in 1999 and 38% in 2016 ($+0.48\%$ year$^{-1}$)[27] (Fig. 1a; Supplementary Table 1). Most of the increase in tree cover and decrease in bare ground/short vegetation occurred after the implementation of forestation policies[18] around the year 2000, making this region the largest tropical area of tree cover increase globally in recent time (Fig. 1b).

Fossil fuel carbon (C) emissions from the eight provinces of southern China amounted to 0.21 Pg C in 1997 and increased to 0.63 Pg C in 2012 where after they stabilized[28] (Fig. 1c). The carbon sequestration in aboveground vegetation was estimated using moderate resolution imaging spectroradiometer (MODIS) and SM and ocean salinity (SMOS) low-frequency passive microwave data (see Methods; refs. [23–25]). We used a static benchmark map of aboveground biomass to train a machine learning algorithm applied on annual MODIS imagery to estimate changes of aboveground carbon density from 2002 to 2017[23], the period in which extensive forest management has taken place in the region. Aboveground biomass retrieved from SMOS-passive microwave observations were calibrated by a set of benchmark maps, which[25] was used as an independent evaluation of changes in carbon density calculated from MODIS optical data

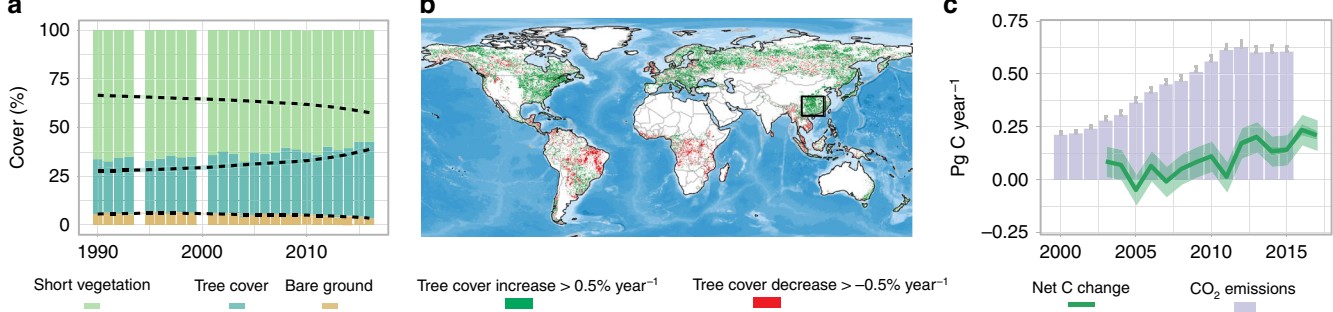

**Fig. 1 Dynamics in forests and carbon stocks in southern China. a** AVHRR estimated[27] dynamics in tree cover, and cover fraction of short vegetation, which is grass and cropland, as well as bare ground from 1990 to 2016 in southern China. **b** The same data source is used to show global increases and decreases in tree cover (larger 0.5% year$^{-1}$) from 2000 to 2016. The study region is marked with a black rectangle. **c** Fossil $CO_2$ emissions (2000–2015)[30] from the eight provinces in southern China (error bars are standard deviations) compared to net C changes (2002–2017) in aboveground vegetation calculated from MODIS data for the same area (shaded area is the uncertainty based on the RMSE between MODIS- and SMOS-based estimates; see text). Pixel number for eight provinces = 8,573,649. Source data are provided as a Source Data file.

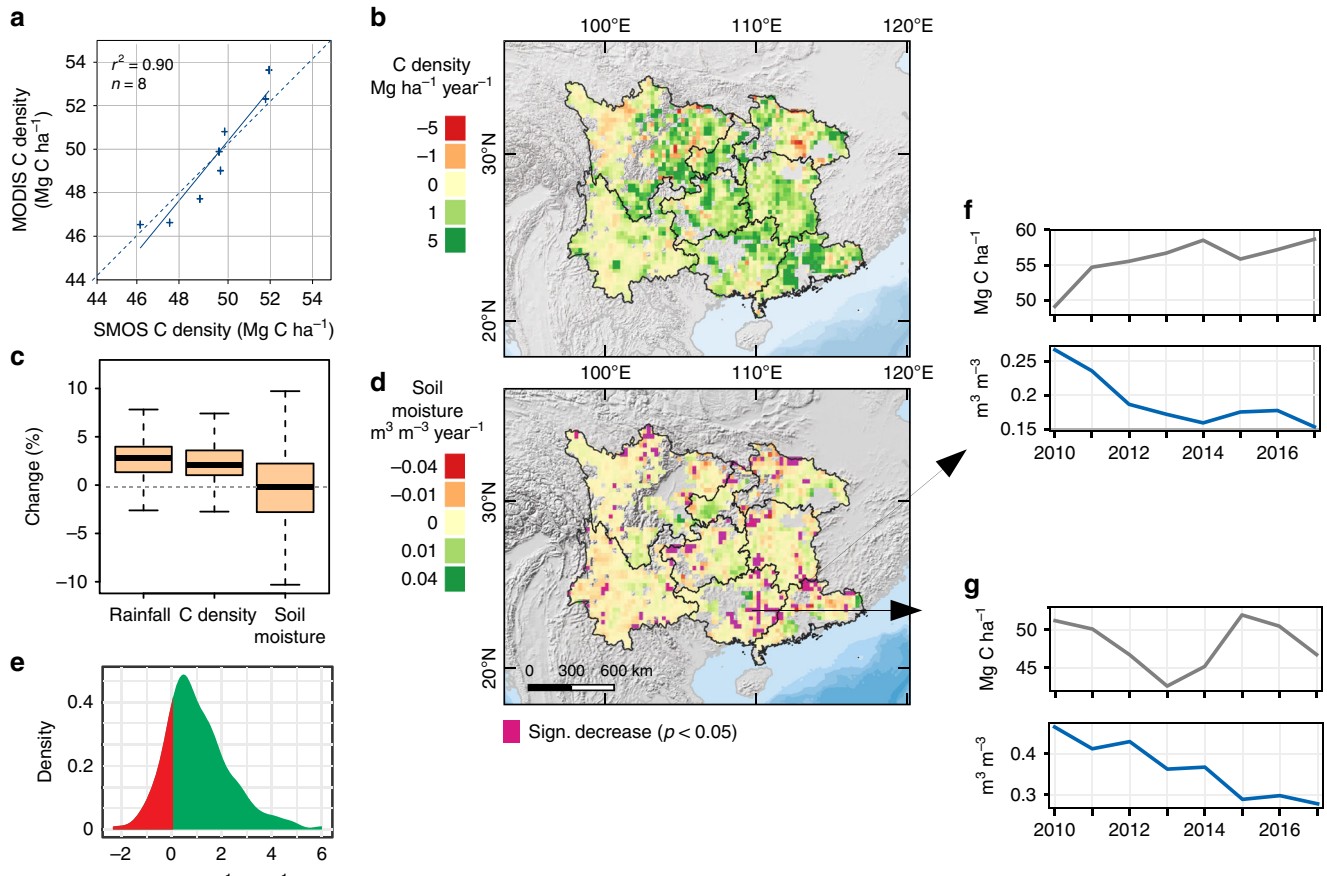

**Fig. 2 SMOS-based carbon density and soil moisture changes 2010–2017. a** Comparison of independent MODIS- and SMOS-based aboveground C density estimates (annual averaged values of the study area). **b** SMOS aboveground C density trends during 2010–2017. **c** Changes (in %) of CHIRPS rainfall, SMOS C density and SMOS soil moisture during 2010–2017. Boxplots show whiskers, quartiles and median across $25 \times 25$ km$^2$ pixels. **d** Spatial distribution of SMOS soil moisture trends during 2010–2017 ($n = 2279$ pixels). Transparent areas have been disregarded for insufficient data quality. Significant negative trends ($p < 0.05$) are shown in purple. **e** Density plot showing the distribution of SMOS aboveground C density trends in the purple areas of **d** with a significant ($p < 0.05$) decrease in soil moisture ($n = 172$). **f** Example of an area with a significant ($p < 0.05$) decrease in soil moisture and a strong increase in C density. **g** Example from Guangxi province showing an area with large-scale tree plantation and massive harvest in 2013 and a significant ($p < 0.05$) decrease in soil moisture.

after the year 2010. The SMOS instrument is able to sense the entire vegetation layer without saturation in densely vegetated areas. The MODIS estimate of the carbon sink was 0.11 (±0.05 temporal uncertainty quantified as root mean square error (RMSE) between MODIS and SMOS) Pg C year$^{-1}$ (0.61 ± 0.2 Mg C ha$^{-1}$ year$^{-1}$) for 2002 to 2017, which represents an amount of carbon equivalent to 20% of the regional annual fossil $CO_2$ emissions from 2002 to 2015, or 33% of the emissions after 2012, a year after which the emissions have stagnated (Fig. 1c). A single drought year, however, can offset this annual mitigation of emissions, as for example in 2011, where the ratio of carbon sink via forest growth to fossil $CO_2$ emissions dropped close to zero.

SMOS low-frequency passive microwave data (available since 2010 at a resolution of $25 \times 25$ km$^2$) showed a high agreement of aboveground carbon changes with the MODIS estimates, both spatially ($r = 0.7$) and temporally ($r = 0.9$) (Fig. 2a, b). The spatial distribution of SM trends derived from SMOS observations (fully independent of the biomass estimations, due to the multi-angular and dual polarization of the sensor) is different from trends in the aboveground biomass C sink during 2010–2017 (Fig. 2c, d). The SMOS SM data showed an overall decrease, while rainfall[29] and biomass C density increased (Fig. 2c). Statistically significant ($p < 0.05$) decreases in SM were found in 8% of the area ($n = 172$ grids with $25 \times 25$ km$^2$); however, the SMOS-based C density trends

were predominantly positive in these areas (Fig. 2e, f). A hotspot of decreased SM was found in Guangxi, which is known for extensive Eucalyptus sp. plantations[30]. Here, the example shown in Fig. 2g points towards a large-scale harvest (followed by regrowth) in 2013.

To further study the impact of different management types on C sinks, we stratified land and forest management over the whole region into eight different types (Table 1), including dense/ minimally managed forest, persistent non-forest, areas with fast/ slow forest gains, deforested areas and two intensities of forest rotation (Table 1; Supplementary Figs. 1–6). The classification is based on mapping of the duration, magnitude and direction of human-induced disturbances (Supplementary Fig. 7) from annual forest probability time-series data (2002–2017, $500 \times 500$ m$^2$ resolution), trained and compared with a full area coverage of 2-m GF-1 satellite imagery. The association of the different land use types with carbon sequestration is shown in Fig. 3, and the spatial distributions are shown in Fig. 4a, b (close-ups are shown in Supplementary Figs. 5, 6). Due to their coarse spatial resolution ($25 \times 25$ km$^2$), biomass carbon change data from SMOS could not be downscaled into the different land use types, but they were used as independent check of the spatially detailed MODIS-based estimates at 500 m resolution. Similarly, SM trends were assessed at 25 km resolution.

**Table 1 Types of land use and land use change in southern China.**

| | Type | Management | Tree plantation/harvest intensity[a] | Tree cover (%)[b] | Forest loss (%)[b] | Mean C density (Mg C ha$^{-1}$)[c] | Net C sink (Pg C year$^{-1}$)/% contribution[c] |
|---|---|---|---|---|---|---|---|
| 1 | Dense forest | Persistent, often protected old forests | Very low/very low | 56 ± 14 | 2.8 ± 7 | 105 ± 11 | 0.005/4% |
| 2 | Forest | Persistent, semi-managed | Low/low | 44 ± 16 | 3 ± 8 | 75 ± 16 | 0.023/20% |
| 3 | Non-forest | Persistent, for example, farmland, sugarcane and fruit trees | Low/low | 14 ± 12 | 0.7 ± 3.4 | 22 ± 16 | 0.032/28% |
| 4 | Recovery | Non-forest to forest (slow) | Medium/none | 28 ± 15 | 0.5 ± 2.5 | 47 ± 12 | 0.016/14% |
| 5 | Afforestation | Non-forest to forest (fast) | Very high/very low | 27 ± 14 | 0.8 ± 3.3 | 44 ± 12 | 0.021/18% |
| 6 | Deforestation | Forest to non-forest | None/very high | 40 ± 16 | 12 ± 17 | 64 ± 13 | −0.00007/−0.06% |
| 7 | Rotation | Medium-scale forestry, changes between forest and non-forest | High/high | 32 ± 16 | 4 ± 10 | 53 ± 11 | 0.017/15% |
| 8 | Rotation$_L$ | Large-scale forestry, low recovery | Low/very high | 37 ± 15 | 9 ± 15 | 61 ± 11 | 0.0008/0.8% |

Dense forest, forest and non-forest are land use types with minor disturbances. The other types include land that experienced changes between forest and non-forest. The percentage of tree cover in each land use type (±standard deviation) is from a Landsat-based tree-cover map in 2010[26]. Forest loss is also derived from[26] and represents the average forest loss per 500-m grid (±standard deviation) from 2000 to 2017. Mean C density and net C sequestration are from our MODIS-based estimates from 2002 to 2017 (±RMSE from calibration biomass data as measure for spatial uncertainty, Supplementary Table 3). See Supplementary Figs. 1–6 for illustrations and Supplementary Table 2 for more details
[a]See Supplementary Fig. 7
[b]Taken from ref. [26]
[c]MODIS estimates 2002–2017 (see Methods)

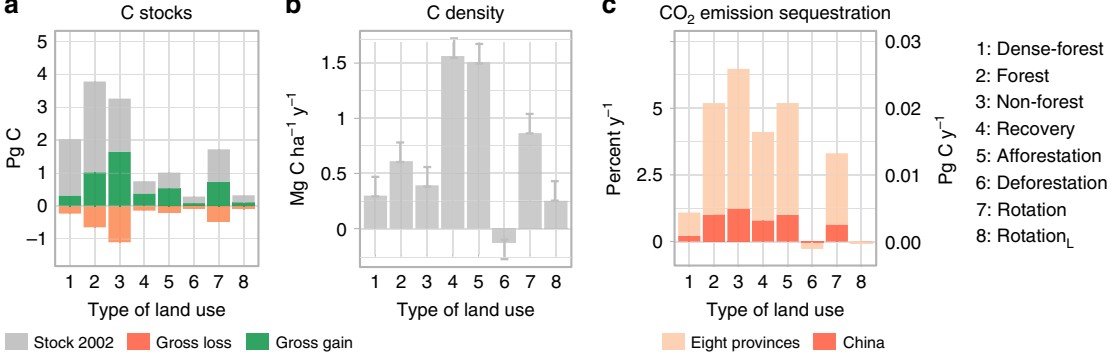

**Fig. 3 Dynamics of carbon stocks for different types of land use. a** C stocks estimated from MODIS data are shown for the first year of study in 2002 in grey, with consecutive gross gains and losses for 2002–2017 in green and orange, respectively. **b** Net changes in C density (2002–2017) from MODIS data. Error bars are the temporal uncertainty based on the RMSE with SMOS estimates. **c** Annual increase in C stocks for each land use type in Pg C year$^{-1}$ on the right-hand axis and as fraction of fossil $CO_2$ emissions for the eight provinces and for China on the left-hand axis (2002–2015). Pixel number for eight provinces = 8,573,649. Source data are provided as a Source Data file.

**Dense forests and persistent forests**. Dense forests are protected remnants of natural forests or dense secondary unharvested forests with a dense tree cover and no disturbances (line 1 in Table 1). Dense forests covered only 8.8% of the region, but stored 20.5% of the total aboveground C stocks (1.71 Pg C) in the beginning of the study period (2002) due to their high C density (105 Mg C ha$^{-1}$ from 2002 to 2017). C dynamics in dense forests remained small during 2002–2017, and contributed 4% (0.005 Pg C year$^{-1}$) to the region's net C uptake, which removed 0.9% of the annual fossil $CO_2$ emissions (Fig. 3a–c).

Persistent forests (20.5% of the area) are less dense forests with minor disturbances, but no major harvests during the observation period (line 2 in Table 1). This land use type stored 2.76 Pg C (33%) in 2002 (mean C density of 75 Mg C ha$^{-1}$) and showed a substantial stock increase (0.023 Pg C year$^{-1}$) from 2002 *to* 2017 (Fig. 3a), contributing 20% to the region's C sequestration, equivalent to 4.2% of the provincial $CO_2$ emissions (Fig. 3c).

**Persistent non-forest land**. Persistent non-forests are typically farmland or grassland, and include small forest patches and fruit trees (line 3 in Table 1). Non-forests covered 43.8% of the region and contained 1.61 Pg C (in 2002), despite a mean C density of 22 Mg C ha$^{-1}$. From 2002 to 2017, the C uptake was 0.032 Pg C year$^{-1}$, caused by the large area occupied (Fig. 3a, b). Net changes in C contributed 28% to the region's C sequestration, storing 5.2% of the annual $CO_2$ emissions (Fig. 3c).

**Recovering forest and afforestation areas**. If an area changed from non-forest to forest, the velocity (the ratio of magnitude to duration) of the change determined the type of forestation (lines 4 and 5 in Table 1). The recovery type is associated with a slow increase in forest cover, likely caused by natural regeneration or planted slow-growing species, and covered 5.4% of the area. In contrast, the afforestation type was associated with a rapid increase in forest area and cover, suggesting plantations of fast-growing species (7.4% of the area). Recovery and afforestation did not include large-scale harvesting, but often included forestation of former agricultural land (Fig. 4c).

The C stocks of these two types of forestation land use were low in 2002 (recovery: 0.37 Pg C; afforestation 0.48 Pg C) (Fig. 3a), but increased rapidly from 2002 to 2017 (recovery: 0.016 Pg C year$^{-1}$; afforestation: 0.021 Pg C year$^{-1}$) (Fig. 3a, b), and contributed 14% (recovery) and 18% (afforestation) to the region's C sequestration, thus storing on average 7.5% of the annual industrial $CO_2$ emissions (Fig. 3c).

**Forest extraction**. We identified three forest land use types involving timber extraction at large scales (lines 6, 7 and 8 in Table 1). The intensity of both the forest removal and replanting defines each type. Conventional (short-)rotation forestry involves clear-cutting of mature stands and subsequent regrowth (i.e., dynamic changes between forest and non-forest) and occurred in

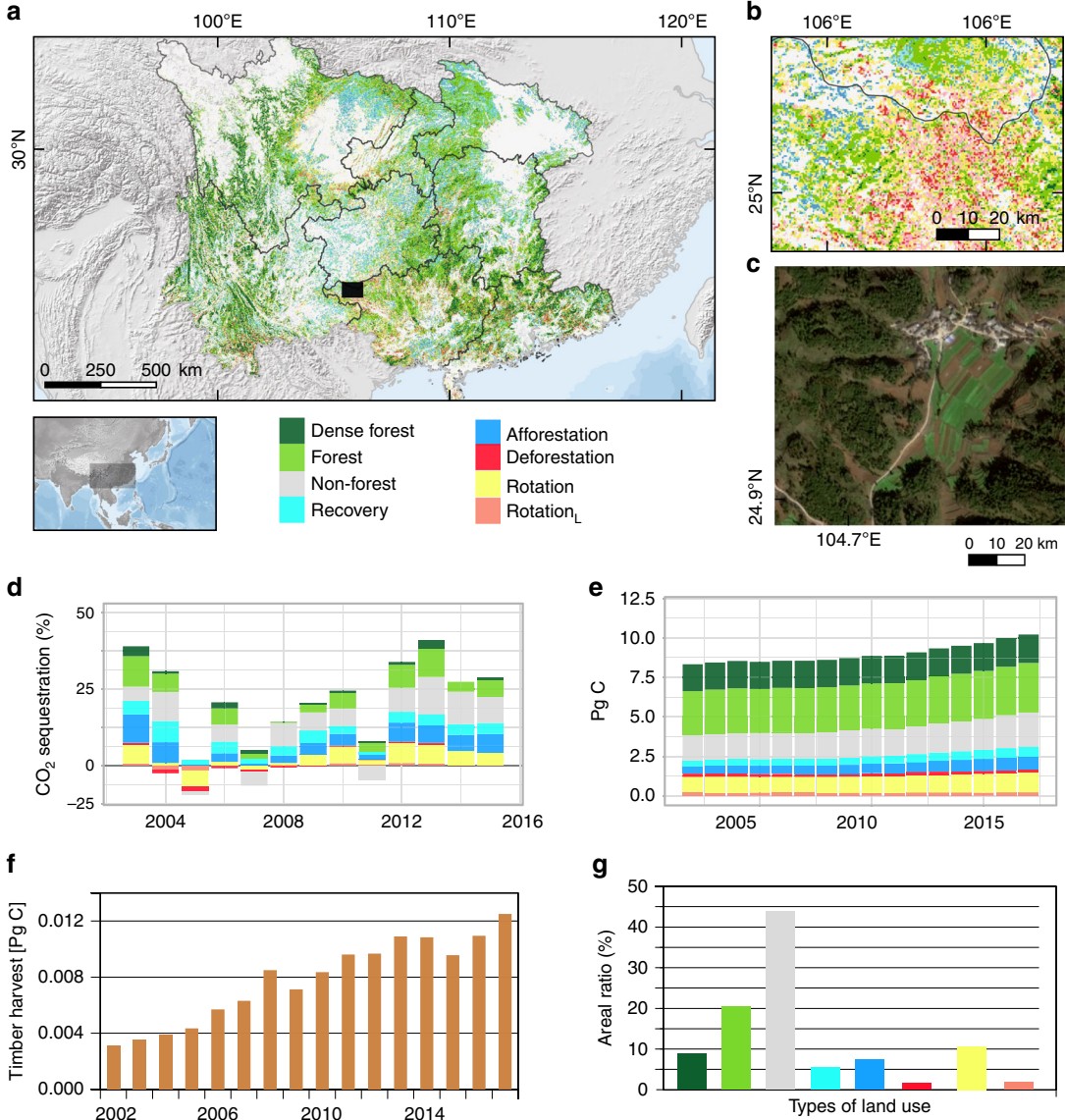

**Fig. 4 Land use in southern China. a** The area classified into the eight types of land use. **b** Forestry is particularly common in northern Guangxi. This close-up is shown as a black rectangle in **a**. **c** Forestation of farmlands on sloping hills is a major reason for the increase in forests. This example image (GF-1, 2-m resolution) is also from northern Guangxi (not the same area as b). **d** Contributions of the types of land use to store fossil $CO_2$ emissions for the eight provinces in southern China (considering net C changes) from 2002 to 2015. **e** C stocks for the types of land use from 2002 to 2017. **f** Annual timber harvest (from forestry statistics at province level; $n = 8$) converted to Pg C. **g** Areal ratios of the types of land use. See Supplementary Figs. 5, 6 for close-up illustrations. Source data are provided as a Source Data file.

10.6% of the area. Harvesting in rotation areas occurred at a rather small spatial scale. Areas that experienced larger-scale harvesting without adequate regrowth were termed rotation$_L$ (1.9% of the area). Forest removal without any regrowth during 2002–2017 was defined as deforestation (1.6% of the area). The C stocks of rotation, rotation$_L$ and deforestation areas were 0.99, 0.22 and 0.2 Pg C in 2002, respectively, with a positive net balance in rotation and rotation$_L$ areas (0.017 and 0.0009 Pg C year$^{-1}$, respectively) and a negative net balance in deforestation areas ($-0.0001$ Pg C year$^{-1}$) between 2002–2017 (Fig. 3a). It has to be noted that the C sink here is calculated from forest growth without considering the full life cycle of extracted wood. It also ignores changes in litter, coarse woody debris and soil carbon. These three land use types with forest extraction make a contribution of 15%, 0.8% and –0.06% to the regional C sequestration and represent an amount of carbon sequestered equivalent to 2.7 % of regional fossil $CO_2$ emissions in rotation areas, whereas

rotation$_L$ and deforestation areas acted as a C source (Fig. 3c) from 2002 to 2015.

Forestry statistics at province level showed that on average 0.008 Pg C were harvested for timber production each year, which amounts to 46% of the average annual net changes (0.017 Pg year$^{-1}$) from *rotation* areas (Fig. 4f). Before 2009, the harvest was higher than the net C sequestration, which however reversed in the later years. Timber harvest accounted for 0.004 Pg C in 2003 and 0.012 Pg C in 2017.

## Discussion

Within the past 20 years the subtropical and mountainous landscapes in southern China have transitioned from traditional agriculture towards managed forests mainly dedicated to the production of wood products. Our study showed that newly planted forests considerably increased C sequestration while more

than tripling timber harvest. The forests that were already present at the beginning of the study period contributed only marginally to the observed large increase in C sequestration, while the plantation of non-forest areas had the highest contribution. Although unharvested forest plantations showed the highest C uptake, harvested forests also had a strongly positive impact on C stocks, as long as harvest was followed by forest regrowth.

A decline in SM in areas of intensive forestation and increased C stocks has been shown by model simulations[10], and the SMOS satellite data indicate a similar development in several areas of southern China. A possible reason is the increased water demand of tree plantations and the associated increased evapotranspiration. Although products on actual evapotranspiration do not agree on the magnitude of increase for the study area (Supplementary Fig. 8), the data support that forest plantations increase evapotranspiration, while lowering the albedo and the land-surface temperature[31,32]. However, while hotspot areas of decreased SM were found to overlay extensive commercial tree plantations (e.g., in Guangxi), statistically significant ($p < 0.05$) decreases in SM were limited to small parts of the area. Moreover, droughts (e.g., in 2011) may have increased the region's evapotranspiration, with implications for the water use efficiency (gross primary productivity/evapotranspiration), which was found to decrease for southern China over the period 2000–2011[33]. The continued monitoring of both C stocks and SM dynamics is thus essential to understand the impact of land use policies and droughts on ecosystems. Here, our study demonstrated the applicability of low-frequency passive microwaves as a tool fulfilling these demands[34,35].

While 30% of the region is covered by protected forests or forests without major active management during the study period, 27% is actively managed as afforestation or from various forest extractive uses. Those managed forests contributed 47% to the region's carbon sequestration, while generating income for the local population through the marketing of wood, other timber products and fruits[19,36]. About one-third of the area's fossil $CO_2$ emissions were stored by aboveground vegetation biomass increase during the past 6 years, and based on this number, it can be extrapolated that an additional forestation area of ~3 million km$^2$ would be required to reach net zero emissions (100%). This is impossible, since only ~1 million km$^2$ are currently non-forested, of which 51% are farmlands[37]. Areas that were not classified as forest also played an important role in our assessment, as the general C uptake in non-forested areas was high. However, the annual C turnover and losses in drought years were considerable in non-forest areas (with only little impact on forests), suggesting limited C sequestration abilities of these areas if tree cover is not increased[38–40].

The spatial extent of *dense forests* (largely old forests) was small, yet our study showed that those forests were an important and stable carbon stock that needs to be preserved. A total of 1.71 Pg C (20.5% of the total C stock) would be lost if all dense forests in southern China were cut, corresponding to ~9 years of fossil $CO_2$ emissions. However, the limited capacity of *dense forests* to store fossil fuel $CO_2$ emissions highlights an inherent dilemma of forest management in relation to mitigating climate change; undisturbed forests maximize the standing C stock and provide a permanent reduction of the atmospheric C content, but once these forests reach a climax state, the net aboveground C sequestration is low[41,42]. Managed forests store less C in the standing biomass and have a smaller potential as permanent C stock, but sequester substantial amounts of C annually[18,42]. To evaluate the potential of managed forest for mitigating climate change, two aspects are critical: first, the harvested areas should be systematically replanted to sustain long-term carbon uptake, and second, the subsequent use of harvested C should be

considered (life-cycle analysis). If the wood enters a short-life product such as paper and swiftly progresses to decomposition or combustion, the C sequestration in the forest is counterbalanced at best or represents a C source when accounting for the C emissions during processing of the wood[43,44]. If the wood is used for construction or other long-term uses, the standing C stock is extended for several decades[45,46]. Finally, using the biomass for bioenergy production while ensuring regrowth following harvest could also help mitigate climate change.

Uncertainties in the present study are mainly related to the spatial resolution of the data sets and the benchmark maps used to calibrate the carbon density maps. The resolution of $25 \times 25$ km$^2$ of the SMOS data is too coarse to distinguish different types of land use. The resolution of $500 \times 500$ m$^2$ of the MODIS data also represents mixtures of different processes, since neither forest expansion/harvesting nor stand increments or degradation usually cover the full extent of a MODIS grid cell. Frequent changes between tree planting and harvesting at small scales pose challenges for accurate classification, which tends to primarily reflect major disturbances measurable at larger scales. Finally, C dynamics estimated with optical data are restricted to the upper green canopy layer, and cannot directly measure the non-green wood part of the vegetation. Larger uncertainties are however mainly found in tropical rainforests[47] with a low spectral variability both in the spatial and temporal domain. This is less of an issue in southern China, where the vegetation is less dense, and rapid canopy changes occur, enabling our machine learning model that translates MODIS optical data into biomass to be trained by observations with a large dynamic range. Moreover, no saturation was found in our carbon estimates (Supplementary Fig. 10), and the relatively good agreement of our assessment based on MODIS and high-resolution land cover data with independent data sets on forest dynamics[28,29] and biomass (SMOS)[27] provide confidence in our estimates (see Methods).

The numbers on C stock increases presented (0.11 Pg C year$^{-1}$ for southern China) are considerably higher than previous estimates, also considering that our numbers are only based on aboveground vegetation biomass. For example, Fang et al.[48] used inventory data to estimate the C sink of forests in the entire China at 0.075 Pg C year$^{-1}$ for the 1980s and 1990s. Piao et al.[38] found the average terrestrial carbon sink of all China (excluding croplands) to be 0.177 Pg C year$^{-1}$ for the same period, of which 58% can be attributed to vegetation biomass. The period of analyses of above-mentioned studies ends, however, around 2003. Here we showed that new forests, which were not present in 2003, contributed with 47% to the region's C sequestration. Moreover, non-forested areas, commonly not considered as a sustainable C sink, contributed further 28%, whereas only 24% of the C sink could be attributed to forests, which were already present in 2003, explaining the large differences to previous studies.

While the increase in C stocks over the period studied is significant, in absolute terms as well as relative to the $CO_2$ emissions of the eight provinces, it is evident that further enhancement of this sink in the future is unlikely. The sloping lands that offer the best opportunity for new forest plantations replacing low productivity croplands are now to a great extent already converted and further increase in C storage will require improved management of currently planted areas to reach a higher C density. Whether this may be seen as a viable option for the region is determined by a range of biophysical, economic and political factors. Further conversions from low to high C density forest types are unlikely, unless economic incentives (e.g., in the form of compensation payments, high market prices, or 'Payments for Environmental Services') are provided. Thus, future sustainable development will notably depend on the extent to which management strategies and economic factors will be able to combine

substantial and increasing C stocks with a favourable economic output. This requires a more refined analysis including different rotation cycles, plantation types (that is species and harvest purpose) and price/demand scenarios.

Although our study found a large increase in C sequestration of which about half was caused by newly planted and managed forests, our numbers also show that storing fossil $CO_2$ emissions from forest plantations alone is not a viable strategy for mitigating anthropogenic carbon emissions, due to the sheer amount of land areas needed. It should be kept in mind, however, that a substantial part of the industrial $CO_2$ emissions in southern China are caused by the production of goods that are not consumed in this area, but are likely nationally and globally traded. Additionally, large-scale forestation programmes can have an adverse impact on the ecosystem by decreasing SM and thus water availability for local organisms, showing that carbon sequestration and economic benefits should not be the only aspects considered when debating the sustainability of land and forest management.

## Methods

**Overview**. This study estimated dynamics in forested areas and C density at a scale of $500 \times 500$ m$^2$ for 2002–2017 (see flowchart in Supplementary Fig. 9). We produced annual forest maps using MODIS, which were compared with independent tree-cover data at scales of 30 m[28] and 5.6 km[27]. We derived a typology of forest continuity and dynamics from the annual forest maps. Since forest management in this area is a very dynamic process, the typology requires annual maps at a reasonable spatial scale, which is not provided by any publicly available product. As analyses were conducted at a spatial scale of 0.25 km$^2$, the land-surface types inevitably included cases of multiple processes and the final classification was determined by the majority of these processes within a grid cell, implying that the data likely over- or underestimated the real areal extents, particularly for forest dynamics. For example, an area of $500 \times 500$ m$^2$ reported to belong to the *afforestation* type means that afforestation was the major process occurring in this grid cell, but does not imply that trees had been planted on the entire area of 0.25 km$^2$.

**MODIS satellite data**. Daily BRDF (bidirectional reflectance distribution function) corrected MODIS imagery (MCD43A4 collection 6)[49] from 2000 to 2017 was the basis for this study. We used all seven bands. Cloud cover is severely impacting the number of good quality observations from optical remote sensing sensors in southern China, and we chose the annual median of daily MCD43A4 images to retrieve an annual image. Due to the fact that even the annual median images were not free of noise, we additionally applied a 3 years moving median window to the annual time series, shortening the available period of data to 2002–2017. This aggressive cleaning was necessary because all kind of noise would be mapped as "clearcut" or other forest dynamic, which would alter our results (see later sections).

**Forest probability**. We created annual forest/non-forest probability maps from annual MODIS data. Here we used forest/non-forest training points manually selected from ~10,000 GF-1 satellite images (pansharpened true-colour composites at $2 \times 2$ m$^2$ for 2013–2017) to train a Random Forest classifier (using the standard setting of 500 trees) with MODIS MCD43A4 reflectance (seven bands) and a Shuttle Radar Topography Mission (SRTM) digital elevation model for predicting the annual forest probability. "Probability" is the standard output of a Random Forest classification and shows how likely a pixel belongs to a certain class. If only two classes are available (here forest and non-forest), the probability shows the likeliness if a pixel belongs to the forest (1) or non-forest (0) class. A total of 110 forest and 323 non-forest training points were manually selected from the GF-1 images, supported by Google Earth. The manual selection of the points followed strict rules: First, the areas should be clearly interpretable in the GF-1 imagery. Second, the areas should be as spatially uniform as possible, that is, either a dense forest or clearly no forest. Third, we used the $500 \times 500$ m$^2$ grid from the MODIS resolution to ensure that a minimum of nine pixels (the point was set in the centre pixel) are covered by the class. Finally, the forest training points should mainly represent dense forests that were stable during 2002–2017 (historical Google Earth images were used when possible), so we used the average of all years as input for the model for each of the seven bands, which further reduced noise and guaranteed a stable model[50]. A three-fold cross-validation of the model (excluding random subsets of the training data that were then predicted) was satisfactory ($r = 0.93$). However, since the chosen points were rather homogeneous, we applied comparisons with independent forest cover maps to assess the quality of the forest probability maps (see section Intercomparison of independent data sets). We applied this model to the annual MODIS bands to generate annual probability maps of the forest/non-forest classification. The probability ranged from 0 to 1 and

indicated the likeliness that an area showed characteristics similar to those of the forest training points, implying that temporally increasing forest probability represented the growth of forests[50]. Likewise, a decrease in forest probability was linked with harvesting. The assumption that forest probability was associated with the fraction of a grid covered by forest was evaluated with independent data (see below).

**Types of land use**. We applied Landsat-based Detection of Trends in Disturbance and Recovery (LandTrendr) to the maps of annual forest probability to identify the types of land uses[51]. We used similar settings as suggested by refs. [52,50] because of comparable data sources and time periods. We set the significance threshold for the model fitting to 0.05 and the maximum number of trajectory segments to six. The LandTrendr algorithm is used for partitioning of the annual data into fitted segments. A segment ends if a major disturbance is detected. The outputs of the algorithm are the number and duration of the segments, and the magnitude of the disturbances. We found no evidence of natural causes of the disturbances to forest probability (only 7% of the disturbances were detected in 2009–2011, when the most severe drought of the past century occurred), suggesting that the disturbances were due to human management. We used the LandTrendr output data to calculate two indices: managed forest increase and managed forest decrease (i.e., forest increases/decreases caused by human management), by dividing the magnitudes of the strongest positive/negative changes by the durations of the changes (Supplementary Fig. 7). The indices ranged from 0 to 1, with high/low values implying rapid and strong/slow and weak changes. A high value in the managed forest increase map was thus interpreted as large-scale plantations of fast-growing trees, and a high value in the managed forest decrease map was interpreted as a clearcut. The indices were classified into several intervals: 0–0.2 (very low), 0.2–0.4 (low), 0.4–0.6 (intermediate), >0.6 (high).

We subsequently used the maps of annual (2002–2017) forest probability to derive nine classes of forest dynamics. The forest classes included (1) dense forests (forest probability ≥0.8), (2) forests (≥0.5) and (3) non-forest (<0.5). The threshold number for dense forests is based on Wang et al. [50], who used field observations to identify old forests (above 0.8). Random Forest commonly uses a probability threshold of 0.5 to distinguish if a pixel belongs to a class or not. We used this threshold to define if an area belongs to the forest (probability above 0.5) or non-forest type. The classes dense forest, forest and non-forest were assigned if the probability scores remained within these intervals during the full period; if not, the type of forest dynamic was determined by the threshold between forest and non-forest (0.5). Class (4) represents forest increases changing from <0.5 in 2002 to >0.5 in 2017 without negative disturbances, and class (5) represents the same but with disturbances. Class (6) represents forest decreases when an area changed from >0.5 in 2002 to <0.5 in 2017 without positive disturbances (e.g., regrowth), and class (7) represents the same but with disturbances. Class (8) represents an area fluctuating between forest and non-forest, but remaining non-forest (2002 compared to 2017), and class (9) represents the same but for remaining forest.

Finally, a grid cell is characterized by three values determining the final land use type: (a) the strongest managed forest increase, (b) the strongest managed forest decrease and (c) the forest dynamic (nine possibilities, previous paragraph). The combination of these three values determined the final type of land use of an area. We also determined the number and duration of the segments for each possible combination to include the velocity and duration of tree growth before harvest to potentially gain insight into the type of tree species (slow- and fast-growing species) (Supplementary Table 2). The final eight types (Table 1; Supplementary Table 2; Supplementary Figs. 2–6) included: (1) dense forests (probability always ≥0.8, no major disturbances), (2) persistent forests (probability always ≥0.5, no major disturbances) and (3) persistent non-forests (probability always <0.5, no major disturbances), which are continuous types with no major human forest management (managed forest index values <0.4) and long-lasting segments (a segment is a period without disturbance). These three types do not exclude planting and harvesting in forest/non-forest areas, but these activities were minor. (4) The recovery class implies a low but positive development in the managed forest index (i.e., no major disturbance, a slow forest increase, no decrease, long-lasting segments) and changes from non-forest to forest. These areas typically had greatly reduced human disturbance throughout the period of analysis that allowed the tree cover to gradually increase. This type does not exclude either tree planting or harvesting in a given grid cell, but the majority of cells had slow and steady increases in tree cover without disturbance. (5) Afforestation represents tree plantations of medium- to fast-growing species and changed rapidly from non-forest to forest. No major decreases were detected (low managed forest index). (6) Deforestation represents areas changed from forest to non-forest, with all segments being negative. (7) Rotation represents areas with frequent changes from non-forest to forest (and vice versa), without the requirement of a particular type as starting and ending years. This type mainly represents the rotation of planting and harvesting activities, which could vary in velocity and magnitude. The average duration of a segment in this type was $4.6 \pm 1.3$ year$^{-1}$, indicating the typical growing cycle of a forest patch[53]. (8) Rotation$_L$ represents fast and rapid decreases in forest probability and changes from forest to non-forest, characterizing areas with large-scale harvesting. The difference between the types of rotation$_L$ and deforestation is that segments of positive slopes can occur in rotation$_L$ areas; however, regrowth does not reach the state of forest (probability <0.5).

**Carbon density**. We used a static unpublished global benchmark map of C density of woody vegetation for 2015 to study forest carbon changes in China[54,55]. The map was constructed using three types of data: (1) more than 8 million waveforms from the Geoscience Laser Altimeter System (GLAS) onboard ICESat-1 (Ice, Cloud and land Elevation Satellite) that reflect the vertical structure of forest and woody vegetation using a semi-systematic sampling approach, (2) a large number of ground data from national inventory and research biomass plots to develop models for converting the GLAS lidar data to C density for different forest types globally and (3) satellite imagery from Landsat, ALOS Phased Array L-band Synthetic Aperture Radar data for 2015, and land elevation data from SRTM. Using GLAS-predicted biomass as the dependent variable, satellite imagery as independent variables and a Bayesian maximum entropy algorithm[54,56], C density and its uncertainty were predicted and were mapped globally at 1-ha spatial grid cells. The machine learning algorithm uses 8 million GLAS derived C density as training data to predict the probability of each pixel falling into a range of C density. The probability maps are then combined to calculate the mean and variance of C density for each pixel[54]. Details of the prediction algorithm and comparisons with other machine learning approaches have been discussed in earlier publications[56]. The map is an improvement over earlier pan-tropical maps by including recent and advanced remote sensing data, and new models that include variations of wood density of trees across tropical regions.

The uncertainty of the map at each grid cell included uncertainty associated with C density models and spatial correlation[54,57]. Although, the previous maps were criticized for dense tropical forests due to use of optical satellite data and the impact of the wood density variations[58], dense rainforests are not present in our study region and no saturation between our data and the benchmark map was observe. While normalized difference vegetation index (NDVI) clearly saturates over forests (Supplementary Fig. 10), satellite imagery from higher-resolution radar sensors provide improved sensitivity to forest structure and biomass across regenerating and lower C density forests across the study region.

The C density map was used to train a model of gradient boosted decision trees with MODIS (MCD43A4 7 bands; NDII, EVI2, MCD43A3 shortwave albedo) and an SRTM elevation model, following[23]. Model training, prediction and validation were done only using data covering the study area. We used 50% of the available pixels ($n = 286,824$) to train the model, and the remaining 50% were used for validation. The quality of the model was generally high ($r = 0.86$, slope = 0.74, RMSE = 11 Mg C ha$^{-1}$), and the C density of the original map could be reproduced (estimates and spatial uncertainty for each class are presented in Supplementary Table 3) without saturation over forests (Supplementary Fig. 10). The model was applied to the data sets for each year to derive maps of annual C density. The dynamics were compared with low-frequency passive microwave satellite data (next section), which also includes an estimation of the temporal uncertainties.

**Intercomparison of independent data sets**. Since no field data on land use types was available, we conducted a visual comparison between the classification and the full areal coverage of GF-1 satellite imagery at a resolution of 2 m (~10,000 images; see Supplementary Figs. 2–4). However, no statistics could be derived from this comparison, thereby requiring additional result comparisons to be taken based on independent data sets. The first step was a comparison with the Landsat-based tree-cover map[26] for 2010, available at a resolution of 30 m (Supplementary Fig. 1). The map was aggregated to 500 m to show the average tree cover of a 500 m pixel from our analysis. The agreement between the aggregated tree-cover map and our forest probability map was very high ($r = 0.88$, $n = 8,796,179$), confirming that our probability maps capture well the average tree cover per grid cell.

We then used the annual long-term data set from ref. [27] for 1982–2016 to provide an overview of the long-term dynamics and to compare with our classification of forest dynamics (Supplementary Fig. 1). The data set is based on long-term data record advanced very high-resolution radiometer (AVHRR) imagery and contains the fractions of tree cover, short vegetation and bare ground at a resolution of 0.05°. The average tree cover (2001–2016) spatially agreed well with our forest probability map (aggregated to 0.05° and averaged over the same period) ($r = 0.81$, $n = 111,954$), confirming that our probability maps could also indicate the fraction of a given grid cell with forests (the higher the probability, the higher the forested fraction). A change in the probability could thus reflect a fractional increase of forest cover within the grid cell. We then compared the temporal dynamics of the tree-cover maps with our classification for the overlapping period (2002–2016). The slopes (increase in forest cover per year) for the forest types (dense forest, forest) (+0.60% year$^{-1}$), non-forest (non-forest) (+0.30% year$^{-1}$) and forest increase (+0.75% year$^{-1}$) (afforestation and recovery) clearly differed (Supplementary Table 1). The forest types (dense forest, forest) had high fractions of forest cover in 1999 (44%) before start of the MODIS time series (as used here). The fraction of forest cover for the forest-increase types started at 26% in 1999, but increased to 41% in 2016. The forest cover for the non-forest type was 14% in 1999 and remained at 21% in 2016.

Deforestation (and replanting) typically occurs at a fine spatial scale, so we applied the Landsat-based forest-loss map from[26] identifying forest losses from 2000 to 2014. We aggregated the 30-m forest-loss map to 500 m and compared the average loss rate (per 500-m grid) with the land use types defined here. The forest losses in the forest, non-forest and forest-increase (afforestation and recovery)

types were very low (Table 1), and only the rotation and deforestation types showed substantial forest losses (Supplementary Fig. 11).

Finally, we compared the MODIS-derived maps of C density with the SMOS L-VOD (L-band vegetation optical depth) data set (currently being the only spatially explicit data set on dynamics of aboveground biomass C)[24,25] as derived from low-frequency passive microwaves. This data set is available at a resolution of $25 \times 25$ km$^2$ for 2010–2017, and the conversion was done using both Saatchi's and Baccini's biomass benchmark maps following[25], and taking the average of both calibrations. At the pixel level, the spatial correlation between the MODIS C density and the L-VOD C density was satisfactory ($r = 0.7$). For temporal comparisons, we averaged for each year both the MODIS C density and L-VOD maps over the region and the temporal dynamics for both data sets agreed well ($r = 0.90$), providing confidence in the dynamics of C density. The uncertainty in the temporal dynamics was assessed by the RMSE between MODIS- and SMOS-derived changes in C density. For an uncertainty assessment based on different biomass calibration maps, we refer to ref. [25]. Due to the high resolution required to distinguish different forest types, the temporal uncertainty could only be provided for the study area as a whole, and the uncertainty of the different forest types was limited to the spatial level. Here we calculated the RMSE between the benchmark map used for calibration and the MODIS-derived C density maps for each forest type.

The negative trend found in SMOS SM is contrasting other studies showing a positive trend in SM this period[24]. However, whereas the SMOS satellite is able to retrieve SM data in densely vegetated areas[59], traditional SM products based on high-frequency passive and active microwave data do not agree in China[60] and often fail in densely vegetated areas[59]. Model simulations and field measurements instead confirm the negative trend in SM[10].

**CO$_2$ emissions**. The data set from ref. [28] contains fossil CO$_2$ emissions for each province for 1997–2015. We used the sectoral approach and converted CO$_2$ to C by dividing the value by 3.67.

**Forestry data**. The timber production data were collected from the Chinese Forestry Statistical Yearbooks, including the timber volumes from 2002 to 2017 for each province. We applied the accumulation method to estimate timber carbon based on the volume of timber. The main species are pine, cedar and eucalyptus, and we used the average wood basic density of these three species to calculate the biomass of the timber production, which was then converted to carbon by multiplying the biomass with 0.5.

**SMOS microwave data**. L-VOD was used as independent data set to assess the dynamics of C density, and SM was used to monitor the patterns of SM dynamics. Both L-VOD and SM products were retrieved simultaneously from the SMOS observations[35]. The use of multi-angular and dual-polarization SMOS observation ensures that L-VOD and SM are independent of each other.

The SMOS L-VOD and SM data sets used here were produced using the SMOS-IC algorithm (version 105) at a resolution of $25 \times 25$ km$^2$ for 2010–2017[61–63]. This algorithm was selected here, as it does not rely on ancillary data such as modelled SM and vegetation optical indices (such as LAI or NDVI), making the SMOS-IC product more robust for applications monitoring both biomass carbon and SM dynamics[62].

SMOS-IC data were processed as follows: first, scene flags were applied and both the L-VOD and SM data for ascending (ASC) and descending (DESC) orbits were filtered according to flags for strong topography, frozen soils (soil temperature <273.5 K), urban areas and water bodies. We then applied a filter for the effects of radio frequency interference (RFI) by excluding data with a TB-RMSE index higher than 10 K. We further applied specific filtering to the L-VOD data: a moving average smoothing (window size = 30 days) was applied on the filtered ASC and DESC L-VOD time series. Outliers, corresponding to data lower (higher) than the 10th (90th) percentile of residues (defined here as the differences between raw L-VOD data and smoothed L-VOD data) were excluded. To further filter out RFI effects, the filtered ASC and DESC L-VOD data were combined into one L-VOD data set by keeping only 30 observations with the lowest TB-RMSE values for each year.

## Data availability

MODIS MCD43 is available from Google Earth Engine. Tree-cover data from Landsat can be downloaded at https://earthenginepartners.appspot.com/science-2013-global-forest/download_v1.2.html. Tree cover from AVHRR (1982–2016) can be downloaded at https://search.earthdata.nasa.gov. CO$_2$ emission data can be downloaded from https://www.nature.com/articles/sdata2017201. SMOS data are available from https://www.catds.fr/. The C density map is available from S.S. The forest management classification and carbon density maps will be made available upon acceptance of the manuscript.

## Code availability

The LandTrendr code can be downloaded at http://emapr.ceoas.oregonstate.edu/tools.html. The IDL version, which was used for this study, is available at https://github.com/KennedyResearch/LandTrendr-2012. Random Forest was run in a GRASS GIS environment with the r.randomforest package.

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

## Acknowledgements

X.T. and Y.Y. were funded by the National Key Research and Development Program of China (no. 2016YFC0502400, 2018YFD1100103), the National Natural Science Foundation of China (41930652), and the Strategic Priority Research Program of Chinese Academy of Sciences (XDA23060100). X.T. was funded by a Marie Curie fellowship, grant number 795970. M.B. received funding from the AXA research grant. R.F. acknowledges the funding from the Danish Council for Independent Research (DFF) Grant ID: DFF-6111-00258. X.X. was funded by the NASA Land Cover and Land Use Change Program (NNX14AD78G). P.C. and J.P. were supported by the European Research Council Synergy grant ERC-2013-726 SyG-610028 IMBALANCE-P.

## Author contributions

X.T. and M.B. contributed equally to this manuscript. X.T., M.B., R.F., M.R.J., S.H., Y.Y., P.C., J.P. and K.W. designed the study. X.T., M.B., Y.Y and Y.W. conducted the analyses with support by P.C., J.P., X.-P.S. and L.F. The data were provided by J.P., L.F., S.S., X.-P.S., B.Z. and C.Z. M.B. and X.T. drafted the manuscript with contributions by P.C., J.P., R.F., K.R., M.R.J., J.-P.W., X.X., X.-P.S., S.H. and all other authors.

## Competing interests

The authors declare no competing interests.
