## [Peer Review File · Nature Communications]

Reviewers' comments:

Reviewer #1 (Remarks to the Author):

This is an interesting paper, that sheds light on the role of forests in mitigating climate change through sequestration of carbon dioxide. It examines this role in a study area in Southern China, where strong changes in forest management turned this region into a substantial carbon sink, which however is facing a wide range of challenges including human and natural factors. I appreciate the way the authors combine remote sensing data with information on AGB to derive annual maps of C density which they use to estimate the amount of C stored in forest biomass. More exciting, however, I find the discussion on the results of this approach, which I find particularly important in the current debate on the potentials of sequestering CO₂. As such, I find the study timely, and well suited for the journal.

However, at the moment I cannot fully recommend the manuscript for publication, as I have several points and comments that I think that need to be addressed to give the readers the chance to fully understand, and possibly reproduce the results.

1. Validation:

It may be just a matter of wording, but in my opinion what the manuscript describes in terms of validation is more what I consider a "result-intercomparison" rather than a pure and independent validation. This fact alone is not a problem by itself, but I think the description needs to be clear - I suggest to revise the manuscript accordingly, both in terms of the remote sensing work as well as the carbon work.

2. Forest Probability mapping:

Here I have a few comments, where I would like to know about some more details:

(1) 110 forest and 323 non-forest plots for the regression appear to me rather low, even when considering the power of machine learning algorithms. What was the reason for these low numbers? In addition, the manuscript does not tell anything about how these points were selected. randomly? manually selected based on image interpretation? What do these points represent: the boundaries of the spectrum (i.e., "no forest at all" and "full canopy"? A revised version of the manuscript should provide more detail on these two issues.

(2) Connected to this: the mode-performance (even though crossvalidated) to me is really interpretable as a measure of "how homogenous are the training data collected?" - unless(!) the points were randomly chosen and labeled.

3. Mapping of land-use types

(1) I like the LandTrendR has been used to segment the time series, but I am missing some information, especially considering the parameterization of the algorithm. to me, this is crucial given that the study used MODIS instead of Landsat (for which landTrendR was originally designed), as well as the (from a forest rotation perspective) short study period.

(2) The "Rotation" class: I am having a hard time imagining how many of these areas are actually strong changes between land-cover types, and not solely caused by errors of the probability output? Is there any way how to give at least an estimate on how good this class is? In addition: I am not an expert for this region of the world, but a full growing cycle of a patch of 4.6yrs to sound extremely short. I may be mistaken here, but I would love to see a reference for this number.

(3) Connected to my comment above: for me, no real validation has been undertaken, but rather a result intercomparison to other datasets which also include uncertainties. I personally have no problem with that, but i suggest being more transparent here - "validation" to me (and possibly to other readers) suggests something different at first.

4. Main manuscript

I find the manuscript well written, and my comments here are only minor.

(1) I find it sometimes confusing that the manuscript reports different time periods. For example, the header in line 83 suggests a time period of analysis between 2002 and 2017, but the first number that is reported is 1982-1999 (line 86). I would suggest to be more clear and more consistent with the reported years, periods, etc.

(2) Also minor: in line 227/228 the manuscript states that "the study shows that the shift in land-use has considerably increased C sinks". Unless I misunderstand something, technically the study did not really test and/or analyzed this, right? I would therefore suggest to modify this statement.

(3) Besides that, a few minor changes for consistency are needed in my opinion (e.g., "land use change" and "land-use change" are not consistently used). After incorporating all changes, I would suggest checking the entire manuscript for consistency.

Reviewer #2 (Remarks to the Author):

The authors have used several remotely sensed data sources and employed a random forest model to classify and detect forest covered areas, and have detected the afforestation/deforestation and carbon stocks across China during 2002-2017. Overall, the paper is well-written, and the figures are high quality and interesting. However, the paper has several minor issues to clarify the methodology and a major concern regarding the classification of old/new forests (see comment number 10). Since the main part of discussion relies on the classification method, such criteria should be revised or justified and better explained (or rephrased) to avoid misunderstanding. It would have not been a big deal for lower impact journals, but I am picky about the phrasing used in Nature publications, as

your conclusions will be soon simplified and used in the community. Please be critical about such a decision and revise it (if necessary).

1. Line 46: "is increasing" >> "are increasing"
2. Figure 1 caption should be revised. "Dynamics in southern China" is not clear.
3. Fig 2a: Do the 8 scatters refer to the long-term average of 8 provinces or the annual values (8 years) of spatially averaged among all provinces?
4. Line 146: "coarse resolution data of SMOS could not be used" >> why not using other finer resolution data products? Is there any finer resolution product available out there besides SMOS? If so, you may consider using that.
5. Lines 235-236: "A possible reason is the increased water demand...">> You may consider calculating Water Use Efficiency (WUE); GPP/ET; or at least discussing it from the literature. Some studies have shown that drought increases ET and affects WUE. Since the study period is a bit short, major drought events may considerably influence ET or water use/demand. Was this the case here? Please discuss about it. Here are a couple of relevant studies (no need to cite them):
 - Ahmadi et al. (2019) "Remote sensing of water use efficiency and terrestrial drought recovery across the contiguous united states." Remote Sensing 11.6: 731.
 - Liu et al. (2015) "Water use efficiency of China's terrestrial ecosystems and responses to drought." Scientific reports 5: 13799.
 - Yu et al. (2017) "Global gross primary productivity and water use efficiency changes under drought stress." Environmental Research Letters 12.1: 014016.
6. Lines 253-254: This sentence is not clear. Please rephrase.
7. Methods, MODIS data: It would be good to mention the exact number of cloud free images that were used in this study for each year.
8. Method, Random Forest: How many trees were chosen for the random forest model? What was the training and testing periods? What were the objective functions (e.g. accuracy, reliability, etc.)?
9. Line 359: "shows likely" >> "shows how likely"
10. Lines 392-394: The classification for old forest versus forests are based on the probability of the machine learning model used. I am wondering why should a higher probability indicate older forest? To me, a higher probability is likely when the vegetation cover is dense, and it does not necessarily indicate the age of forests. Perhaps NDVI of leaf area index would have been useful measures to include in the machine learning model for detecting how dense the vegetation is. Since most of the results and discussions are based on this subjective classification (why 0.8 and 0.5? why not 0.85?), I am not sure how can one rely on them.
11. Line 442: "C density and its uncertainty were predicted" >> How? This needs to be explained.

Reviewer #3 (Remarks to the Author):

Review: Forest management in southern China generates an extensive carbon sequestration

General Comments: This is a very well written manuscript with a critical message about climate change mitigation and the role forests can (and have) played in China. I particularly like the soil moisture implications and I think there needs to be more on this topic in the future.

There are some items that need to be addressed before publication.

Title: The title is a little awkward (as if a word is missing). How about: Forest management in southern China generates extensive carbon sequestration benefits

Abstract: It is unclear (at this point in reading) how/why the harvested forest is contributing to the sequestration benefits? Is this just through NEP or is there a wood product pool associated with the accounting? In which case, the boundaries of the carbon dynamics over time (when did harvest start?) get harder to implement.

How are you accounting for respiration and net C uptake?

Because you are only measuring aboveground tree C change (not total ecosystem C uptake and release), this does not equal CO₂ offsets as stated here: Our study demonstrates that intensive land management in southern China is offsetting ~30% of regional fossil CO₂ emissions during the last 6 years. I do not understand how you are calculating the offset when only net aboveground growth is only being considered. You could state that X amount of CO₂ had been removed and stored, but without accounting for the C released by biological or human processes (harvest and wood products) you cannot state a 1:1 offset.

Introduction:

What do you mean by “sustainable forest management practices” in this sentence: Sustainable forest harvesting practices can maintain or increase standing carbon stocks and at the same time generate economic output from timber products.

There is very little evidence that harvest of any kind either maintains or increases carbon stocks compared to no-harvest scenarios (Harmon et al. 1990, Stockmann et al. 2012, Williams et al. 2016, Hudiburg et al. 2019). If you mean ‘sustainable’ as compared to more severe harvest practices could you please clarify this?

Results:

This seems like a very important result: “A single drought year, however, can offset the annual mitigation of emissions as in 2011, where the ratio of carbon sink via forest growth to fossil CO₂ emissions dropped close to zero.” I assume this is reflected in the Net C change in Figure 1 C? What happened in 2005 and 2007?

Thank you for defining the forest management types so specifically!

Line 196: Forest extraction. It needs to be noted that the C ‘sink’ is simply from regrowth and does not account for the fate of removed wood nor the residency time of that carbon before it returns to the atmosphere (generally much quicker than if not harvested). Unless you are subtracting the carbon harvested in your equation?

Discussion:

Line 269: Yes, LCA is critical, but you have not done that here?

Line 276: "If the wood is used for construction or other long-term uses, the standing C stock is extended and can be considered as a permanent stock". This is simply not true. Even the longest half-lives observed and used in LCA average 100 years (Skog 2008, Dymond 2012) and this is just for single family housing (the longest lived structures) in North America. All other construction has much shorter half lives. I do not know much about China's construction or wood product industry but I do not think the construction or products would be longer-lived.

References: There are several references where the formatting needs to be fixed.

Suggested References to include:

Dymond, C. C. 2012. Forest carbon in North America: annual storage and emissions from British Columbia's harvest, 1965–2065. *Carbon Balance and Management* 7:8-8.

Harmon, M. E., W. K. Ferrell, and J. F. Franklin. 1990. Effects on carbon storage of conversion of old-growth forests to young forests. *Science* 247:699-702.

Hudiburg, T. W., B. E. Law, W. R. Moomaw, M. E. Harmon, and J. E. Stenzel. 2019. Meeting regional GHG reduction targets requires accounting for all forest sector emissions. *Environmental Research Letters* 14:095005.

Skog, K. E. 2008. Sequestration of carbon in harvested wood products for the United States. *Forest products journal*. Vol. 58, no. 6 (June 2008): Pages 56-72.

Stockmann, K. D., N. M. Anderson, K. E. Skog, S. P. Healey, D. R. Loeffler, G. Jones, and J. F. Morrison. 2012. Estimates of carbon stored in harvested wood products from the United States forest service northern region, 1906-2010. *Carbon Balance and Management* 7:1.

Williams, C. A., H. Gu, R. MacLean, J. G. Masek, and G. J. Collatz. 2016. Disturbance and the carbon balance of US forests: A quantitative review of impacts from harvests, fires, insects, and droughts. *Global and Planetary Change* 143:66-80.

Reviewer #1 (Remarks to the Author):

This is an interesting paper, that sheds light on the role of forests in mitigating climate change through sequestration of carbon dioxide. It examines this role in a study area in Southern China, where strong changes in forest management turned this region into a substantial carbon sink, which however is facing a wide range of challenges including human and natural factors. I appreciate the way the authors combine remote sensing data with information on AGB to derive annual maps of C density which they use to estimate the amount of C stored in forest biomass. More exciting, however, I find the discussion on the results of this approach, which I find particularly important in the current debate on the potentials of sequestering CO₂. As such, I find the study timely, and well suited for the journal.

Authors: We thank the reviewer for the support of our study!

However, at the moment I cannot fully recommend the manuscript for publication, as I have several points and comments that I think that need to be addressed to give the readers the chance to fully understand, and possibly reproduce the results.

1. Validation:

It may be just a matter of wording, but in my opinion what the manuscript describes in terms of validation is more what I consider a "result-intercomparison" rather than a pure and independent validation. This fact alone is not a problem by itself, but I think the description needs to be clear - I suggest to revise the manuscript accordingly, both in terms of the remote sensing work as well as the carbon work.

Authors: We fully agree, the term "validation" may be misleading in this context. We have reworded the manuscript as suggested and replaced the term "validation" with "comparison" and similar terms. The word "validation" is now only used in relation to the decision tree model.

2. Forest Probability mapping:

Here I have a few comments, where I would like to know about some more details:

(1) 110 forest and 323 non-forest plots for the regression appear to me rather low, even when considering the power of machine learning algorithms. What was the reason for these low numbers? In addition, the manuscript does not tell anything about how these points were selected. randomly? manually selected based on image interpretation? What do these points represent: the boundaries of the spectrum (i.e., "no forest at all" and "full canopy"? A revised version of the manuscript should provide more detail on these two issues.

Authors: We agree that 433 points does not sound much, however, they represent "pure" classes to give a best possible range for the probability (0-1), and pure classes at this resolution (500 x 500 m) are rarely found, are generally homogeneous, and are labor intensive to map.

We agree that this was not sufficiently described and have added the following text in line 337: "A total of 110 forest and 323 non-forest training points were manually selected from the GF-1 images, supported by Google Earth. The manual selection of the points followed strict rules: First, the areas should be clearly interpretable in the GF-1 imagery. Second, the areas should be as spatially uniform as possible, i.e. either a dense forest or clearly no forest. Third, we used the 500 x 500 m grid from the MODIS resolution to ensure that a minimum of nine pixels (the point was set in the centre pixel) are covered by the class. Finally, the forest training points should mainly represent dense forests that were stable during 2002–2017 (historical Google Earth images were used when possible)".

(2) Connected to this: the mode-performance (even though crossvalidated) to me is really interpretable as a

measure of "how homogenous are the training data collected?" - unless(!) the points were randomly chosen and labeled.

Authors: That is correct, and here we agree that these cross validated numbers have to be read with care, as the reviewer correctly notes. This is why we have chosen additional independent data sets for inter-comparison of the result maps, which are the forest cover maps by Hansen and Song.

We have added this important information to the text, line 386: "*However since the chosen points were rather homogeneous, we applied comparisons with independent forest cover maps to assess the quality of the forest probability maps (see section inter-comparison of independent data sets).*"

3. Mapping of land-use types

(1) I like the LandTrendR has been used to segment the time series, but I am missing some information, especially considering the parameterization of the algorithm. to me, this is crucial given that the study used MODIS instead of Landsat (for which LandTrendR was originally designed), as well as the (from a forest rotation perspective) short study period.

Authors: We have added the requested parameters to the text, line 397: "*We used similar settings as suggested by Sulla-Menashe 2014 and Wang et al, 2018 because of comparable data sources and time periods. We set the significance threshold for the model fitting to 0.05 and the maximum number of trajectory segments to six.*"

(2) The "Rotation" class: I am having a hard time imagining how many of these areas are actually strong changes between land-cover types, and not solely caused by errors of the probability output? Is there any way how to give at least an estimate on how good this class is? In addition: I am not an expert for this region of the world, but a full growing cycle of a patch of 4.6yrs to sound extremely short. I may be mistaken here, but I would love to see a reference for this number.

Authors: Thanks. We have put a lot of effort in getting the MODIS data as robust and error free as possible, applying several filters that should remove sudden drops caused by bad data. Also the LandTrendR applies another fitting, so the final time series should be quite robust. Of course, we cannot fully exclude that pixels end in the rotation class because of noise. For us, the average segment number of 4.6 years was the ultimate evidence because the harvest period of Eucalyptus, which is by far the most planted species in this area, is about 6 years. We have added a reference to support this here. Another independent number giving confidence in the class comes from the forestry statistics: the timber harvest of the 8 provinces for the same time period is 0.008 Pg C per year, and the net changes from the rotation class is 0.017, so the harvest is about 50%, which makes sense.

Li, X. *et al.* Effects of Successive Rotation Regimes on Carbon Stocks in Eucalyptus Plantations in Subtropical China Measured over a Full Rotation. *PLOS ONE* **10**, e0132858 (2015).

(3) Connected to my comment above: for me, no real validation has been undertaken, but rather a result intercomparison to other datasets which also include uncertainties. I personally have no problem with that, but I suggest being more transparent here - "validation" to me (and possibly to other readers) suggests something different at first.

Authors: We agree and do not use the term validation any more.

4. Main manuscript

I find the manuscript well written, and my comments here are only minor.

(1) I find it sometimes confusing that the manuscript reports different time periods. For example, the

header in line 83 suggests a time period of analysis between 2002 and 2017, but the first number that is reported is 1982-1999 (line 86). I would suggest to be more clear and more consistent with the reported years, periods, etc.

Authors: Unfortunately, the different data sets have different temporal extents, so we cannot fully avoid reporting different periods. The period 1982–1999 is used to illustrate how the area looked like before the forest management programs were put in action. The study period is 2002–2017. We have made this clearer in the revised version of the text.

(2) Also minor: in line 227/228 the manuscript states that "the study shows that the shift in land-use has considerably increased C sinks". Unless I misunderstand something, technically the study did not really test and/or analyzed this, right? I would therefore suggest to modify this statement.

Authors: Thanks. We reformulated the sentence, line 241: "*Our study showed that newly planted forests have considerably increased C sequestration while more than tripling timber harvest*".

(3) Besides that, a few minor changes for consistency are needed in my opinion (e.g., "land use change" and "land-use change" are not consistently used). After incorporating all changes, I would suggest checking the entire manuscript for consistency.

Authors: Thanks, we have revised this and checked the manuscript carefully.

Reviewer #2 (Remarks to the Author):

The authors have used several remotely sensed data sources and employed a random forest model to classify and detect forest covered areas, and have detected the afforestation/deforestation and carbon stocks across China during 2002-2017. Overall, the paper is well-written, and the figures are high quality and interesting. However, the paper has several minor issues to clarify the methodology and a major concern regarding the classification of old/new forests (see comment number 10). Since the main part of discussion relies on the classification method, such criteria should be revised or justified and better explained (or rephrased) to avoid misunderstanding. It would have not been a big deal for lower impact journals, but I am picky about the phrasing used in Nature publications, as your conclusions will be soon simplified and used in the community. Please be critical about such a decision and revise it (if necessary).

Authors: We thank the reviewer for the positive words, the suggested revisions and the support! We agree that at this level of journal we need to be precise with formulations and we have thus changed the formulation of old-forest to dense forest.

1. Line 46: "is increasing" >> "are increasing"

Authors: Corrected.

2. Figure 1 caption should be revised. "Dynamics in southern China" is not clear.

Authors: We name it now "*Dynamics in forests and carbon stocks in southern China*".

3. Fig 2a: Do the 8 scatters refer to the long-term average of 8 provinces or the annual values (8 years) of spatially averaged among all provinces?

Authors: These are the annual values among all provinces, we have added this information to the caption.

4. Line 146: “coarse resolution data of SMOS could not be used” >> why not using other finer resolution data products? Is there any finer resolution product available out there besides SMOS? If so, you may consider using that.

Authors: Unfortunately, the SMOS data are the only available low frequency passive microwave observations (apart from SMAP, from which the temporal coverage is however too short). As such, it is a unique data set and no finer resolution data can compare with them. The coarse resolution is due to the functioning of the sensor, which received passive emissions from the vegetation. See also our newest global study: <https://www.nature.com/articles/s41477-019-0478-9>.

5. Lines 235-236: “A possible reason is the increased water demand...”>> You may consider calculating Water Use Efficiency (WUE); GPP/ET; or at least discussing it from the literature. Some studies have shown that drought increases ET and affects WUE. Since the study period is a bit short, major drought events may considerably influence ET or water use/demand. Was this the case here? Please discuss about it. Here are a couple of relevant studies (no need to cite them):

- Ahmadi et al. (2019) "Remote sensing of water use efficiency and terrestrial drought recovery across the contiguous united states." Remote Sensing 11.6: 731.
- Liu et al. (2015) "Water use efficiency of China's terrestrial ecosystems and responses to drought." Scientific reports 5: 13799.
- Yu et al. (2017) "Global gross primary productivity and water use efficiency changes under drought stress." Environmental Research Letters 12.1: 014016.

Authors: Thank you for this interesting suggestion, which we are happy to follow. Since the study includes already many different data sets, we decided to discuss this point in the discussion part. We have added the following text, line 255: “*Moreover, droughts (for example in 2011) may have increased the region's evapotranspiration, with implications for the water use efficiency (gross primary productivity/evapotranspiration) which was found to decrease for southern China over the period 2000–2011 (Liu et al., 2015)*”.

6. Lines 253-254: This sentence is not clear. Please rephrase.

Authors: We have removed most part of the sentence, it reads now “*This is impossible, since only ~ 1 million km² are currently non-forested, of which 51% are farmlands³⁶*”.

7. Methods, MODIS data: It would be good to mention the exact number of cloud free images that were used in this study for each year.

Authors: This information is difficult to provide, since we started with annual images where each pixel was the median of daily observations. Since the median was used, clouds were largely excluded (they are extreme values). The total number of images used would thus be 365 per year, however merged to a median image.

8. Method, Random Forest: How many trees were chosen for the random forest model? What was the training and testing periods? What were the objective functions (e.g. accuracy, reliability, etc.)?

Authors: We have added the requested information to the text. The number of trees is included now in 500 and the other information can now be found in the same paragraph.

9. Line 359: “shows likely” >> “shows how likely”

Authors: Corrected.

10. Lines 392-394: The classification for old forest versus forests are based on the probability of the machine learning model used. I am wondering why should a higher probability indicate older forest? To me, a higher probability is likely when the vegetation cover is dense, and it does not necessarily indicate the age of forests. Perhaps NDVI of leaf area index would have been useful measures to include in the machine learning model for detecting how dense the vegetation is. Since most of the results and discussions are based on this subjective classification (why 0.8 and 0.5? why not 0.85?), I am not sure how can one rely on them.

Authors: Thanks. We fully agree that the density of the forest does not necessarily relate to its age. To be 100% accurate, we changed the term to “dense forest” instead of old-forest. We had chosen the term after selecting the training points for the forest probability classification by studying forests in Google Earth and the GF-1 high resolution satellite data, in association with local knowledge of our group. So the higher the probability, the closer is the pixel to the selected training points, which are presumably chosen in old-forest areas, but we agree it is preferable to be accurate in terminology and now term it “dense forest”. We also agree that it would have been a good idea to include NDVI in the model, however, the inclusion of the red and near infrared bands should lead to a comparable result. The selection of the threshold values was based on Wang et al., 2018, who used field observations from the Loess Plateau (bordering our study area) to set the number for old-forests. The commonly used standard setting of the RandomForest model uses a probability threshold of 50% to distinguish if a pixel belongs to a class or not. Here, we used this setting to define if a pixel ranging from 0 to 100 belongs to the forest (probability above 50%) or nonforest class. We make this now clear in the text, line 416: “*The threshold number for dense forests is based on⁵⁰, who used field observations to identify old-forests (above 0.8). Random Forest commonly uses a probability threshold of 0.5 to distinguish if a pixel belongs to a class or not. We used this threshold to define if an area belongs to the forest (probability above 0.5) or non-forest type.*”.

11. Line 442: “C density and its uncertainty were predicted” >> How? This needs to be explained.

Authors: We have added more information here, line 469:”*The machine learning algorithm uses 8 million GLAS derived C density as training data to predict the probability of each pixel falling into a range of C density. The probability maps are then combined to calculate the mean and variance of C density for each pixel. Details of the prediction algorithm and comparisons with other machine learning approaches have been discussed in earlier publications*”

Reviewer #3 (Remarks to the Author):

Review: Forest management in southern China generates an extensive carbon sequestration
General Comments: This is a very well written manuscript with a critical message about climate change mitigation and the role forests can (and have) played in China. I particularly like the soil moisture implications and I think there needs to be more on this topic in the future.
There are some items that need to be addressed before publication.

Authors: We thank the reviewer for the support of our study. We are pleased that the soil moisture part is perceived well!

Title: The title is a little awkward (as if a word is missing). How about: Forest management in southern China generates extensive carbon sequestration benefits

Authors: Thank you for the suggestion, we think the problem came from the “an” and we have changed the title to: *Forest management in southern China generates extensive carbon sequestration*”.

Abstract: It is unclear (at this point in reading) how/why the harvested forest is contributing to the sequestration benefits? Is this just through NEP or is there a wood product pool associated with the accounting? In which case, the boundaries of the carbon dynamics over time (when did harvest start?) get harder to implement.

Authors: Thanks for spotting this missing information. Here we do not account for wood product pools but only assess NEP. It is an assessment of the annual aboveground C sink in vegetation biomass of areas which are classified as “rotation”, i.e. areas where forest harvest takes place. Soil carbon changes are not included neither. We have added this information: *“Forest growth in harvested forest areas contributed 16% and non-forest areas 28%, while timber harvest was tripled.”*

How are you accounting for respiration and net C uptake?

Because you are only measuring aboveground tree C change (not total ecosystem C uptake and release), this does not equal CO₂ offsets as stated here: Our study demonstrates that intensive land management in southern China is offsetting ~30% of regional fossil CO₂ emissions during the last 6 years. I do not understand how you are calculating the offset when only net aboveground growth is only being considered. You could state that X amount of CO₂ had been removed and stored, but without accounting for the C released by biological or human processes (harvest and wood products) you cannot state a 1:1 offset.

Authors: We agree that the way we have formulated these sentences can be misleading and the word “offset” is not correct. We have replaced “offset” consistently throughout the manuscript by “store” or “remove an amount equivalent to”.

Introduction:

What do you mean by “sustainable forest management practices” in this sentence: Sustainable forest harvesting practices can maintain or increase standing carbon stocks and at the same time generate economic output from timber products.

There is very little evidence that harvest of any kind either maintains or increases carbon stocks compared to no-harvest scenarios (Harmon et al. 1990, Stockmann et al. 2012, Williams et al. 2016, Hudiburg et al. 2019). If you mean ‘sustainable’ as compared to more severe harvest practices could you please clarify this?

Authors: Reviewer is right; the meaning of the sentence was unclear. What we want to emphasize is that a moderate forest harvesting as compared to farmlands without any forest can generate C sequestration and income. We changed the sentence accordingly, line 54: *“Moderate forest harvesting practices on forested farmlands can generate carbon stocks and at the same time provide economic output from timber products”*.

Results:

This seems like a very important result: “A single drought year, however, can offset the annual mitigation of emissions as in 2011, where the ratio of carbon sink via forest growth to fossil CO₂ emissions dropped close to zero. “ I assume this is reflected in the Net C change in Figure 1 C? What happened in 2005 and 2007?

Authors: Yes, it refers to the net C change Figure 1c. The year 2011 was used in the text as an example year, because it was a historically severe drought in this area, which does not exclude other drought years (as likely 2005 and 2007). We have made it clear now in the text that 2011 represents a year of severe drought in this area, thereby serving as a showcase for the impact of drought.

Thank you for defining the forest management types so specifically!

Line 196: Forest extraction. It needs to be noted that the C 'sink' is simply from regrowth and does not account for the fate of removed wood nor the residency time of that carbon before it returns to the atmosphere (generally much quicker than if not harvested). Unless you are subtracting the carbon harvested in your equation?

Authors: Thanks, reviewer is right, the C sink comes from regrowth and does not account for the fate of removed wood. We have clarified this in the text, line 218: *"It has to be noted that the C sink here is calculated from forest growth without considering the full life cycle of extracted wood. It also ignores changes in litter, coarse woody debris and soil carbon."*

Discussion:

Line 269: Yes, LCA is critical, but you have not done that here?

Authors: LCA was not part of this study but only discussed in the discussion. We agree it is a very critical topic, which however cannot be assessed from a remote sensing perspective (which is the major perspective of this study).

Line 276: "If the wood is used for construction or other long-term uses, the standing C stock is extended and can be considered as a permanent stock". This is simply not true. Even the longest half-lives observed and used in LCA average 100 years (Skog 2008, Dymond 2012) and this is just for single family housing (the longest lived structures) in North America. All other construction has much shorter half lives. I do not know much about China's construction or wood product industry but I do not think the construction or products would be longer-lived.

Authors: Thanks for highlighting this statement, which we agree is in fact wrong. We have reformulated the sentence as follows, line 290: *"If the wood is used for construction or other long-term uses, the standing C stock is extended for several decades (Dymond, 2012)"*.

References: There are several references where the formatting needs to be fixed.

Authors: Thanks, we have now fixed the formatting.

Suggested References to include:

- Dymond, C. C. 2012. Forest carbon in North America: annual storage and emissions from British Columbia's harvest, 1965–2065. *Carbon Balance and Management* 7:8-8.
- Harmon, M. E., W. K. Ferrell, and J. F. Franklin. 1990. Effects on carbon storage of conversion of old-growth forests to young forests. *Science* 247:699-702.
- Hudiburg, T. W., B. E. Law, W. R. Moomaw, M. E. Harmon, and J. E. Stenzel. 2019. Meeting regional GHG reduction targets requires accounting for all forest sector emissions. *Environmental Research Letters* 14:095005.
- Skog, K. E. 2008. Sequestration of carbon in harvested wood products for the United States. *Forest products journal*. Vol. 58, no. 6 (June 2008): Pages 56-72.
- Stockmann, K. D., N. M. Anderson, K. E. Skog, S. P. Healey, D. R. Loeffler, G. Jones, and J. F. Morrison.

2012. Estimates of carbon stored in harvested wood products from the United States forest service northern region, 1906-2010. Carbon Balance and Management 7:1.

Williams, C. A., H. Gu, R. MacLean, J. G. Masek, and G. J. Collatz. 2016. Disturbance and the carbon balance of US forests: A quantitative review of impacts from harvests, fires, insects, and droughts. Global and Planetary Change 143:66-80.

Authors: Thanks for the suggestions; we have included several of them in the revised version of the manuscript.

REVIEWERS' COMMENTS:

Reviewer #1 (Remarks to the Author):

Dear author team,

I think you have done a great job in addressing my comments and concerns, as well as those made by the other two reviewers. In my opinion the manuscript has improved a lot, and I am happy to say that I can recommend the manuscript for publication.

Reviewer #2 (Remarks to the Author):

Thank you for point by point response to all the comments. My suggestions have been addressed in the revised manuscript, and the paper should be ready for publication.

Reviewer #3 (Remarks to the Author):

Thank you for addressing my comments so thoroughly. Excellent manuscript! I recommend publication.